# Formulation of Alkali-Activated Slag Binder Destined for Use in Developing Countries

**Nabil Bella** [1] , **Edwin Gudiel** [2] , **Lourdes Soriano** [3], **Alba Font** [3] , **María Victoria Borrachero** [3], **Jordi Paya** [3] **and José Maria Monzó** [3,*]

1   FIMAS Laboratory, Tahri Mohamed University of Bechar, Bechar 08000, Algeria; bella.nabil@univ-bechar.dz
2   Departamento de Arquitectura y Urbanismo, Universidad Andina del Cusco, Cusco 08006, Peru; egudiel@uandina.edu.pe
3   Grupo de Investigación en Química de los Materiales (GIQUIMA), Instituto de Ciencia y Tecnología del Hormigón (ICITECH), Universitat Politècnica de València (UPV), 46022 València, Spain; lousomar@upvnet.upv.es (L.S.); alprefon@upvnet.upv.es (A.F.); vborrachero@cst.upv.es (M.V.B.); jjpaya@cst.upv.es (J.P.)
*   Correspondence: jmmonzo@cst.upv.es

**Abstract:** Worldwide cement production is around 4.2 billion tons, and the fabrication of one ton of ordinary Portland cement emits around 900 kg of $CO_2$. Blast furnace slag (BFS) is a byproduct used to produce alkali-activated materials (AAM). BFS production was estimated at about 350 million tons in 2018, and the BFS reuse rate in construction materials of developing countries is low. AAM can reduce $CO_2$ emissions in relation to Portland cement materials: Its use in construction would be a golden opportunity for developing countries in forthcoming decades. The present research aims to formulate AAM destined for future applications in developing countries. Two activators were used: NaOH, $Na_2CO_3$, and a mixture of both. The results showed that compressive strengths within the 42–56 MPa range after 28 curing days were obtained for the $Na_2CO_3$-activated mortars. The characterization analysis confirmed the presence of hydrotalcite, carbonated phases, CSH and CASH. The economic study showed that $Na_2CO_3$ was the cheapest activator in terms of the relative cost per ton and MPa of manufactured mortars. Finally, the environmental benefits of mortars based on this reagent were evidenced, and, in terms of $kgCO_2$ emissions per ton and MPa, the mortars with $Na_2CO_3$ yielded 50% lower values than with NaOH.

**Keywords:** alkali-activated material; blast furnace slag; activators; mechanical properties; microstructure; developing countries

## 1. Introduction

Worldwide cement production in 2019 was 4.2 billion metric tons, while China alone produced 2.2 billion metric tons and India 320 million metric tons. Overall cement production is estimated to increase from 3.27 billion metric tons in 2010 to 4.83 billion metric tons by 2030 [1]. Cement production is responsible for about 8% of the anthropogenic $CO_2$ emissions [2]. The Intergovernmental Panel on Climate Change (IPCC) proposes that global anthropogenic $CO_2$ emissions need to lower by 45% from 2010 by 2030 to limit global warming to 1.5 °C [3]. The use of alkali-activated materials (AAM) can reduce $CO_2$ emissions up to 80% compared to Portland cements materials by using blast furnace slag (BFS) or fly ash (FA), among other precursors [4].

In 2018, the steel industry's worldwide slag production was around 600 million tons, of which 58% was BFS, and 42% was steelworks slag [5]. Recycling BFS in developed countries (e.g., Germany) is nearly 100%, but utilization rates differ from country-to-country [5]. In developed countries,

changing construction procedures to more innovative processes is very difficult because the inertia of the construction industry is set, and there is no growing market, unlike developing countries. This represents a golden opportunity to develop a more ecological construction industry in developing countries by increasing BFS use in construction and reducing $CO_2$ emissions by lowering Portland cement production. The use of BFS for AAM applied to structures being used is already a reality in countries like Australia [6].

Alkali-activated slag (AAS) as cementing material was first investigated in the 1930s by Khul, a German researcher [7]. Later, AAS was further studied in the former Soviet Union by V.D. Glukhovskii in the 1950s, with the first application in 1958, which was formally patented in 1974. Many eastern European countries and China undertook many projects with AAS (road slabs, air-field runways and precasting) in the 1970s and 1980s with excellent durability [8,9]. In the 1980s, the term geopolymer was introduced by Davidovits [10], who first mixed alkali with a calcined mixture of kaolinite, limestone and dolomite. In the 1990s, when environmental concerns surged, interest in AAM increased because these materials can reduce $CO_2$ emissions [4,11,12]. In 1996, Shi published one of the first scientific papers to assess the strength, porosity and permeability of AAS mortar activated with different activators (NaOH, $Na_2SiO_3$ and $Na_2CO_3$) [13]. This author concluded that the mortars activated with $Na_2SiO_3$ presented the highest strength, the lowest porosity and the finest pore structure. Fernández-Jiménez et al. [14] studied the influence of various factors, such as specific slag surface, curing temperature, activator concentration and the nature of activators. They concluded that an alkaline nature was the most significant factor in obtaining better mechanical strength. The best-employed activator as a mixture of $Na_2SiO_3$ and NaOH to obtain mortars with compressive strength values over 100 MPa and flexural strength at around 12 MPa.

BFS was classified as a moderately calcium-rich material that can be activated under relatively moderate alkaline conditions [15]. The main reaction products are gels CSH (calcium silicate hydrate) and CASH (calcium silico-aluminate hydrate), with the minor formation of hydrotalcite and some zeolites like gismondine [15,16]. Generally speaking, the most widely used activators are NaOH and $Na_2SiO_3$, or a mixture of both. For the BFS system, optimal pH values are around 13–13.6, and the addition of $Na_2SiO_3$ normally causes greater development of mechanical strength compared to those systems activated with only NaOH. The silicate source reacts with $Ca^{+2}$ from dissolved slag to form denser products [16].

The use of $Na_2CO_3$ instead of other activators can be an environment-friendly and economical alternative for developing countries because it is 2–3 times cheaper than $Na_2SiO_3$ and NaOH [17]. Awoyera and Andesina [18] made a review on the application of alkali-activated slag and also considered $Na_2CO_3$ the most appropriate activator. However, the long setting time and/or slow strength development make their use less appealing than the other activators [17]. Combining $Na_2CO_3$ with other activators can improve AAS performance. For example, the use of MgO and $Na_2CO_3$ mixture improves early strength and drying shrinkage in relation to those mortars that use only $Na_2CO_3$ as an activator [19]. Ke et al. [20] studied the effectiveness of employing calcined-layered double hydroxides to control the kinetics reaction of slags activated with $Na_2CO_3$. These authors concluded that the addition of this reagent resulted in a faster setting and a better hardening process. Kovtun et al. [21] explored the use of NaOH, Portland cement, and a mixture of silica fume and slaked lime as accelerators for systems BFS-$Na_2CO_3$. Employing accelerators brought about an enhancement of 12 MPa in the mortars cured for 48 h vs. the control mortar.

The majority of recent AAS research works have been conducted in developed countries rather than developing AAS to be used in developing countries with specifications of high-level technology and many material resources, especially chemical reagents. This paper aims to formulate an AAS that adapts to the specificities of developing countries, which exhibits rapid urbanization and considers local human and materials resources and existing infrastructures; for example, Sub-Saharan Africa. The ONU memberships formulated the Sustainable Development Agenda and included 17 Sustainable Development Goals (SDG) to protect the planet and to eradicate poverty [22]. SDG 11 proposes

making cities inclusive, safe, resilient and sustainable, while SDG 13 is about climate action and claims regulating $CO_2$ emissions. To this end, AAS was formulated to be used in precasting as blocks or tiles that take into account the specifications of developing countries.

The present paper investigates employing NaOH, $Na_2CO_3$, and a mixture of both to prepare mortars with BFS as a precursor and preparing AAM to be utilized in developing countries.

## 2. Materials and Methods

The BFS herein used was provided by Cementval SL (Puerto de Sagunto-Valencia, Spain). BFS was dry-milled in a ball mill for 30 min (450 g of BFS were milled with 98 alumina balls). The commercial reagents (NaOH and $Na_2CO_3$) were supplied by PANREAC QUIMICA SLU (Barcelona, Spain).

Some characteristics of the used BFS are detailed using techniques like XRF (X-ray fluorescence) or FESEM (field emission scanning electron microscopy).

The chemical composition of BFS was measured in a Magix Pro spectrometer (Philips) by the XRF technique. Table 1 provides the oxide composition per weight percent (wt %).

**Table 1.** Chemical composition of blast furnace slag (BFS) (wt %).

| $Al_2O_3$ | $SiO_2$ | CaO | $Fe_2O_3$ | $K_2O$ | MgO | $Na_2O$ | $SO_3$ | Other | * LOI |
|---|---|---|---|---|---|---|---|---|---|
| 10.60 | 30.04 | 40.35 | 1.30 | 0.57 | 7.47 | 0.87 | 1.94 | 1.30 | 5.56 |

* LOI—loss of ignition determined at 950 °C for 1 h.

The equipment used to measure granulometric distribution was a Mastersizer 2000 (Malvern Instruments, Malvern, UK). The technique followed by the equipment was laser diffraction. BFS was dispersed in water and stirred at 1200 rpm by applying a 1-min ultrasound before measurements. The milled BFS presented a unimodal curve with a narrower shoulder of particles below 1 µm (Figure 1).

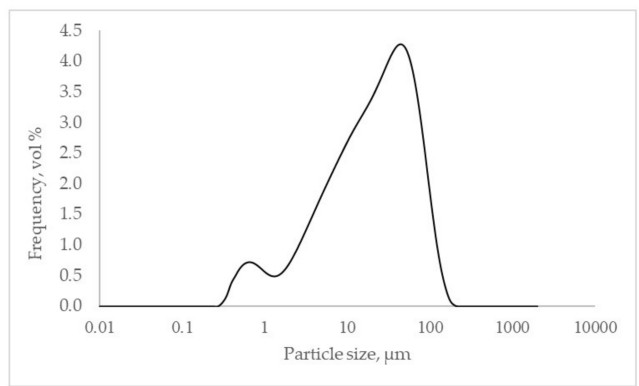

**Figure 1.** Granulometric distribution of milled BFS.

Brucker AXS D8 (Billerica, MA, USA) Advance equipment was employed to carry out X-ray diffraction (XRD) analyses. The range went from 5 to 70 2θ degrees using Cu Kα radiation at 20 mA and 40 kV, with a 2 s accumulation time and a 0.02 angle step. The XRD diffractogram of BFS is represented in Figure 2. The halo located within 20–35 2θ degrees was attributed to the amorphous phase of BFS. A crystalline phase attributed to the presence of calcite (C, $CaCO_3$, PDF#050586) can also be observed.

Finally, the morphology of the milled BFS was observed by FESEM. The equipment employed was ULTRA 55 ZEISS equipment. BFS was carbon-coated to observe it, and the image was taken at 2 kV. The micrograph of the BFS particle morphology is represented in Figure 3. The milled BFS presented an irregular morphology with different size particles as a result of the milling process.

Two types of alkali activators were used: NaOH, $Na_2CO_3$, and a mixture of both. The samples activated with only NaOH were called SH, and the chosen molarities were 3, 4 and 5 M. The samples activated with $Na_2CO_3$ were named SC, and the chosen molarities were 1.5, 2, 2.5, 3 and 3.5 M. Finally,

the samples that used the mixture of two reagents were called SHC, and the chosen molarities differed for both products. For example, the mixture named SHC 2/1 had molarity of 2 M for NaOH and a molarity of 1 M for $Na_2CO_3$. All the combinations of activators are represented in Table 2. In this research, mortars were prepared at a BFS/sand weight ratio of 1:3 (450:1350 g) and at a BFS/water ratio of 0.4 (180 g of $H_2O$). The humidity of the natural siliceous sand (from the company Caolines Lapiedra, Liria, Valencia, Spain) was less than 1% with a specific weight of $2.71 \pm 0.15$ g/cm$^3$. Mortars were cast in $4 \times 4 \times 16$ cm$^3$ molds.

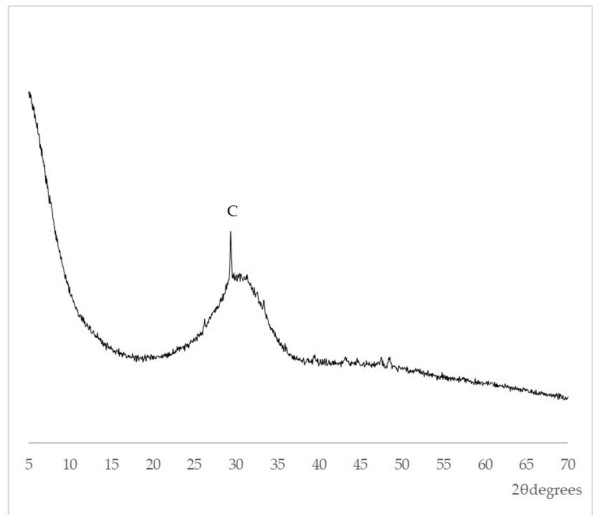

**Figure 2.** XRD diffractogram of BFS.

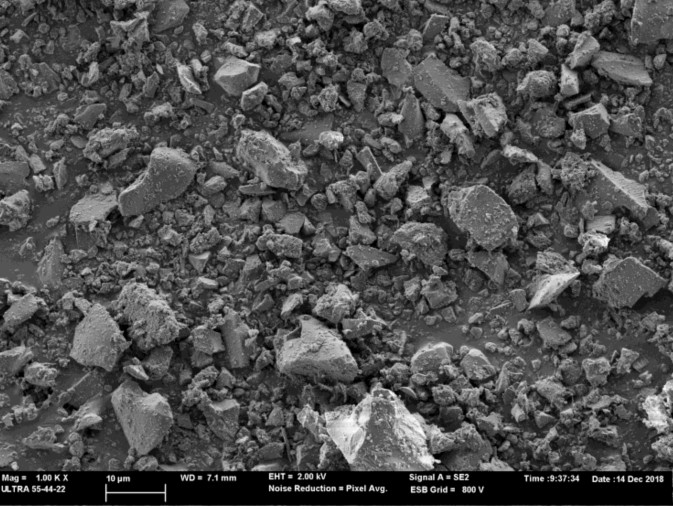

**Figure 3.** FESEM micrograph of the milled BFS at 1000× magnification.

Mortars were cured in a conditioned chamber (at 20 °C, 100% RH), demolded after 24 h, weighed and returned to the conditioned chamber until mechanical testing commenced (7, 28 and 90 curing days), which was performed in a universal Instron Model 3382 machine with the charge rate set at 1 mm/min according to EN 196-1 [23].

Microstructural analyses were done only on some selected pastes (without sand) cured under the same conditions as mortars. On the testing day, the hardened paste was ground, submerged in acetone to stop hydration reactions, and were filtered and dried for 30 min at 65 °C before being passed through a 125 μm sieve. For the microstructural analysis of XRD, thermogravimetry (TG) and Fourier-transform infrared spectroscopy (FTIR) was performed. The XRD and FESEM equipment are previously described. The TG analysis was run in a Mettler Toledo TGA 850 module. Samples were

placed in alumina crucibles with sealed lids and heated from 50 °C until 1000 °C at a heating rate of 20 °C/min in an open-air atmosphere. The FTIR analysis was conducted in a Brucker model Tensor 27 spectrometer (medium infrared rank from 7800 to 370 cm$^{-1}$). In the present work, the analysis rank went from 2000 to 400 cm$^{-1}$.

**Table 2.** Doses of alkali dissolutions for mortars.

| Names | Molarity NaOH | Molarity Na$_2$CO$_3$ | Moles of Na$_2$O * | NaOH (g) * | Na$_2$CO$_3$ (g) * |
|---|---|---|---|---|---|
| SH 3 | 3.0 | – | 0.27 | 21.6 | – |
| SH 4 | 4.0 | – | 0.36 | 28.8 | – |
| SH 5 | 5.0 | – | 0.45 | 36.0 | – |
| SC 1.5 | – | 1.5 | 0.27 | – | 28.6 |
| SC 2 | – | 2.0 | 0.36 | – | 38.2 |
| SC 2.5 | – | 2.5 | 0.45 | – | 47.7 |
| SC 3 | – | 3.0 | 0.54 | – | 57.2 |
| SC 3.5 | – | 3.5 | 0.63 | – | 66.8 |
| SHC 3/0.5 | 3.0 | 0.5 | 0.36 | 21.6 | 9.5 |
| SHC 2/1 | 2.0 | 1.0 | 0.36 | 14.4 | 19.1 |
| SHC 1/1.5 | 1.0 | 1.5 | 0.36 | 7.2 | 28.6 |

* These quantities were selected for a mixture of 1350 g of sand, 450 g of BFS and 180 g of water.

As both the economic and environmental aspects play one of the most important roles for the present research work, a detailed study was conducted. The cost and kgCO$_2$ emissions of the activators per ton of the prepared mortar (€/kg), and mechanical behavior (€/ton × MPa), were determined. The results obtained for the selected assessed mortars allowed us to compare the use of the different activators herein studied.

## 3. Results

### 3.1. Mechanical Strength

Figures 4 and 5 represent the flexural (Rf) and compressive (Rc) strengths of all the mortars studied for 7, 28 and 90 days curing, respectively.

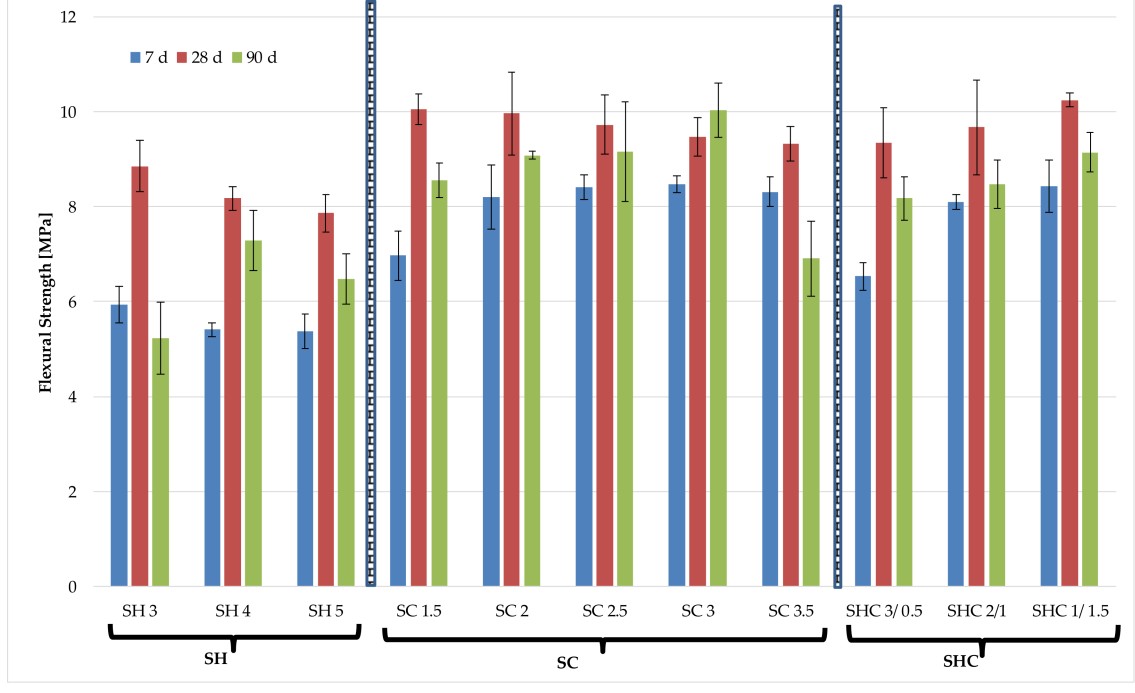

**Figure 4.** Flexural strength (Rf) of mortars for 3, 28 and 90 curing days.

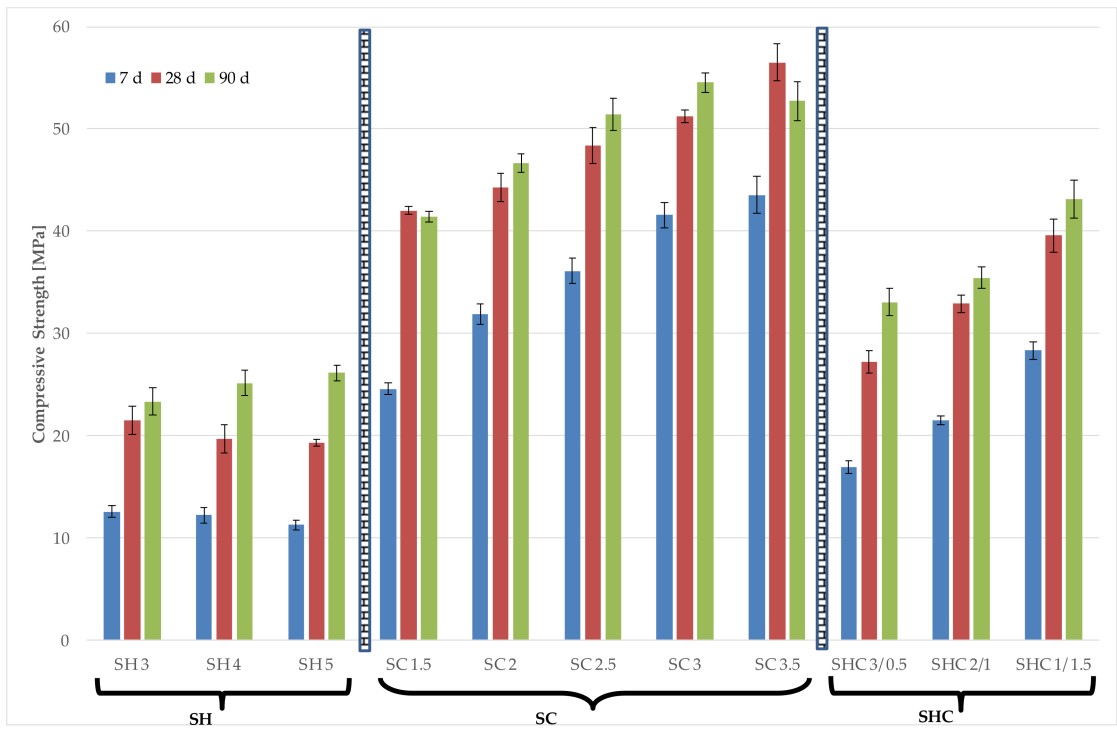

**Figure 5.** Compressive strength (Rc) of mortars for 3, 28 and 90 curing days.

After analyzing the flexural strength results separately for the three studied mixtures, namely NaOH (SH series), $Na_2CO_3$ (SC series), and NaOH and $Na_2CO_3$ mixtures (SHC series), it highlights the following results: (a) for NaOH, an increase in Rf took place between 7 and 28 curing days. For the 90 days curing time, Rf decreased with increased activator molarity; (b) for $Na_2CO_3$ as an activator after 7 curing days, all the mortars had similar flexural strength values, except for the mortar with the lower quantity activator. After 28 curing days, all the values were similar, and strength slightly decreased in all the mortars after 90 curing days except for the mortar SC3; (c) when the NaOH and $Na_2CO_3$ mixture was used as the activator, the mortar with the lesser $Na_2CO_3$ quantity obtained the lower strength value at 7 curing days, but no significant difference among the three types of mortars was observed after 28 and 90 curing days. The flexural strengths after 28 curing days of the mortars were higher than after 90 curing days. The decreased flexural strength from 28 days to 90 days was attributed to the shrinkage cracks, in line with what Pacheco-Torgal et al. [24] suggested. Many of the mortars with the three different activators reached a flexural strength between 8 and 10 MPa.

When the three activator mixtures types were compared to one another, it concluded that the mortars containing $Na_2CO_3$ had higher flexural strength than those activated with only NaOH.

Figure 5 represents the Rc of all the studied mortars. The obtained results showed that $Na_2CO_3$ was the activator that yielded the highest Rc values. After studying the results for each tested activator, we highlight the following observations:

a.    When NaOH is used, for the studied molarities range, there was no global correlation between the molarity of the NaOH solution and the Rc for the three studied curing days. An increase in Rc was observed with curing time. For any mortar, the compressive strength at 90 days exceeded 30 MPa. Wang et al. [25] reported for NaOH that variation in the quantity of NaOH within 3–11% of the range (per slag weight) had very little effect on strength. In the present study, the NaOH percentages varied between 4.8% (SH3) and 8% (SH5). For the wider variations in the NaOH concentration, other authors obtained significant differences in compressive strength. For example, de Mores Pinheiro et al. [26] made mortars with BFS activated with NaOH and KOH: for the NaOH-activated mortars, those activated with 4 M of NaOH (6.4% per slag weight) at

　　　65 °C for 7 days obtained a compressive strength of 17.47 MPa compared to 31.11 MPa obtained by the mortar activated with 12 M of NaOH (19.2% per slag weight).

b.　　The $Na_2CO_3$-activated mortars displayed much better mechanical performance than the corresponding NaOH-activated ones. The Rc values at 90 curing days fell within the 40–55 MPa range; that is, about twice that for NaOH. An increase in Rc was observed when increasing the quantity of $Na_2CO_3$ as the activator (SC 2.5, SC 3 and SC 3.5). These results are consistent with other authors, such as Yuan et al. [27], who obtained the best compressive strengths for the mortars with the biggest $Na_2CO_3$ quantity (analyzed range between 3 and 5% for $Na_2O$ wt %). It is remarkable that for 7 curing days, the Rc values were higher than those for the NaOH-activated mortars after 90 days. This means that the activation rate of slag by $Na_2CO_3$ was high, and good early age mechanical performance was achieved. The maximum Rc for 7 curing days corresponded to 3.5 M the concentration of $Na_2CO_3$ (SC 3.5), and this value was higher than 40 MPa.

c.　　When the NaOH and $Na_2CO_3$ mixtures were used as activating reagents, Rc was enhanced when the quantity of $Na_2CO_3$ in the mixture was increased. Although all the mortars of SHC the series had the same quantity of $Na_2O$ in relation to BFS weight, the best results were obtained for the mortars with the largest quantity of $Na_2CO_3$ (SHC 1/1.5). Collins and Sanjayan [28] demonstrated the benefits of using $Na_2CO_3$ in combination with NaOH, but we observed that the NaOH/$Na_2CO_3$ mixture was much less efficient in terms of compressive strength development. In all cases, Rc was lower than the values obtained in the series with only $Na_2CO_3$.

　　　In order to compare the role of both the activator's nature and concentration, a 28-day curing time for compressive strength versus the percentage in weight of $Na_2O$ to respect slag is plotted in Figure 6. The range of the studied $Na_2O$ percentages went from 3.72% to 8.68%.

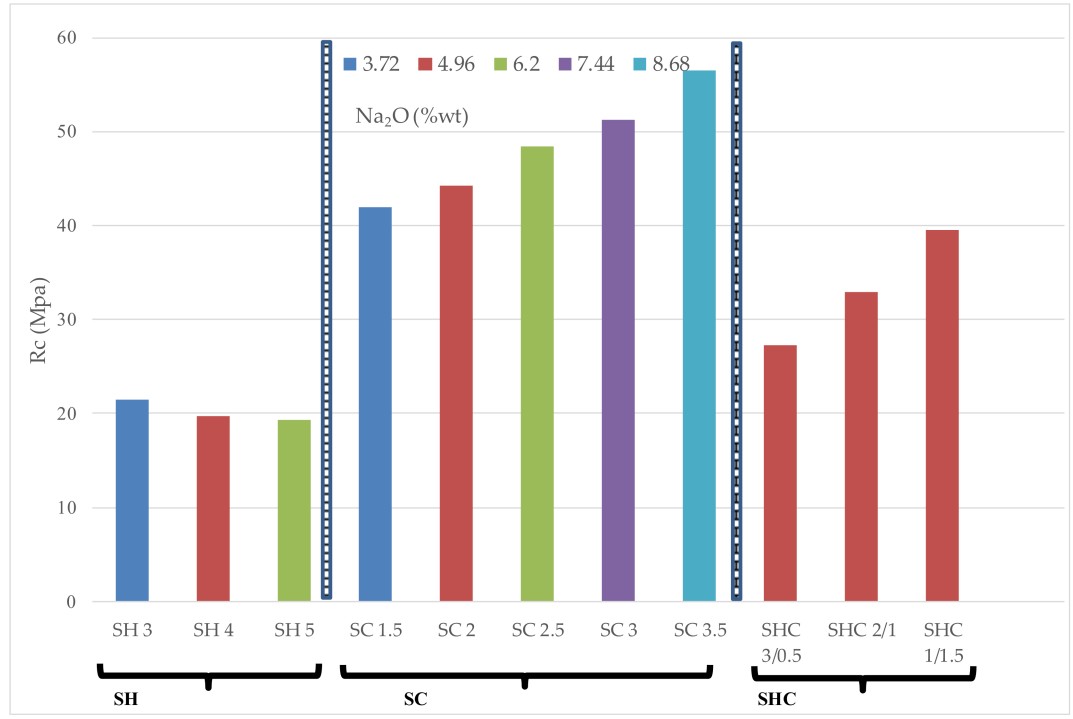

**Figure 6.** Compressive strength of mortars (cured for 28 days) according to $Na_2O$ percentage in relation to BFS content.

　　　The importance of anion nature in the compressive strength development for a given $Na_2O$ content was demonstrated: the mortars activated with carbonate had higher compressive strengths than those activated with hydroxide. This was observed for all the $Na_2O$ contents studied in the

present study. Li et al. [29], Fernandez-Jiménez and Puertas [30], and Bernal et al. [31] assumed that the anion group in alkaline solution was the dominant factor. The mixtures activated with $Na_2CO_3$ have a lower pH in a solution, and this factor can affect the setting time and strength development, but only at early ages (the first four days). Moreover, at mid- and long-term ages, pH values do not negatively affect strength development.

Although some authors suggest an optimum $Na_2O$ range within the range from 3% to 5% [14], it was demonstrated that this range depends on the activator's nature: with NaOH, no significant differences were found for the studied $Na_2O$ contents, but the best results were obtained by the highest $Na_2O$ percentage for $Na_2CO_3$.

Jiao et al. [32] stated that the $OH^-$ ion dissolves the glassy structure of BFS. When $Na_2CO_3$ was added, the $OH^-$ concentration lowered. However, the highly cross-linked structure of the carbonated compounds enhanced compressive strength.

## 3.2. Thermogravimetric Analysis (TGA/DTG)

To simplify the microstructural analysis, only a specific $Na_2O$ percentage was chosen. The chosen pastes were SH 4, SC 2 and SHC 1/1.5. They all had a $Na_2O$ weight percentage of 4.96% in relation to the BFS. Paste SHC 1/1.5 was chosen because it had the highest mechanical strength of its series. The DTG curves at the different curing ages are represented in Figure 7. On the DTG curves, three mass loss intervals were selected: (a) the first interval was below 300 °C and showed two peaks. Peak 1 can be attributed to the loss of the combined water of CSH gel and peak 2 to the loss of the combined water of the CASH gel [19,33,34]. The main peak was the CSH gel; (b) the second was 300–500 °C, in which peak 3 developed (attributed to the decomposition of the hydrotalcite phase [19,33]); (c) the last interval was between 500 and 1000 °C (it contained peaks 4 and 5) and was attributed to carbonate phases like calcium and sodium carbonates or carboaluminates [19].

As shown in Figure 7, peaks 1, 2 and 3 appeared at practically all the curing time and for all the activators, but with different heights and width peaks. Peaks 4 and 5 were more visible for series SC and SHC that contained $Na_2CO_3$ as an activator. The pastes with $CO_3^{-2}$ ions produce compounds of type carboaluminate ($C_3A.CaCO_3.12H_2O$) [20]. The partial and total mass losses for all the studied pastes are summarized in Table 3 for the different curing ages. The mass loss in the interval 50–300 °C presented a great evolution between 7 and 28 curing days; these data confirm that the most developed products within this time interval are CSH and CASH gel.

The study of mass loss within different temperature intervals corroborated the more marked presence of the carbonated phases in pastes SC and SHC. The mass loss derived from the hydrotalcite phase was greater in paste SH. The total mass loss increased with curing time and demonstrated the hydration reaction evolution.

**Table 3.** Mass loss as a percentage for the thermogravimetry (TG) curves for the pastes cured for 7, 28 and 90 curing days.

| Paste | Age (Days) | % Mass Loss 50–300 °C | % Mass Loss 300–500 °C | % Mass Loss 500–1000 °C | % Mass Loss 50–1000 °C |
|---|---|---|---|---|---|
| SH 4 | 7 | 9.88 | 1.89 | 1.34 | 13.11 |
| | 28 | 12.19 | 2.60 | 2.03 | 16.82 |
| | 90 | 12.25 | 3.12 | 2.10 | 17.47 |
| SC 2 | 7 | 8.72 | 2.06 | 3.81 | 14.59 |
| | 28 | 12.94 | 2.29 | 3.73 | 19.10 |
| | 90 | 12.04 | 2.73 | 4.59 | 19.36 |
| SHC 1/1.5 | 7 | 7.83 | 2.31 | 3.42 | 13.56 |
| | 28 | 13.98 | 2.46 | 3.32 | 19.80 |
| | 90 | 12.15 | 3.02 | 4.46 | 19.64 |

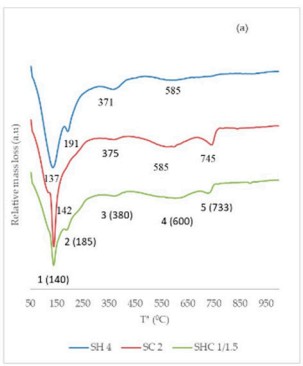
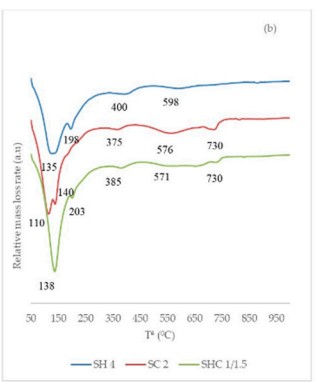

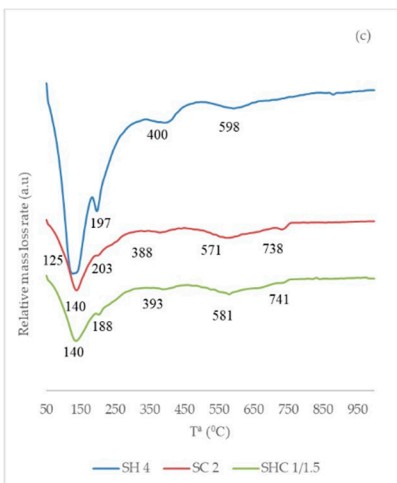

**Figure 7.** DTG curves of the pastes cured for: (**a**) 7 days; (**b**) 28 days; (**c**) 90 days.

### 3.3. FTIR Analysis of Pastes

Figure 8 shows the FTIR spectra of the pastes cured at 7, 28 and 90 days at ambient temperature. Table 4 presents the main assignations of the vibrations on the FTIR curves. The assignment was based on some research works by different authors [35–38].

The unreacted BFS presented various peaks: (a) at around 1477 $cm^{-1}$, the peak was identified as the asymmetric stretching mode of O-C-O in $CO_3^{-2}$ due to the presence of calcite; (b) the broadband centered at 933 $cm^{-1}$ corresponded to the asymmetric stretching vibration mode of Si-O-T (T: Si or Al); (c) the shoulder at 875 $cm^{-1}$ was attributed to the asymmetric stretching of the $AlO_4$ groups; (d) the peak at 663 $cm^{-1}$ corresponded to vibration modes of sulfate, (e) finally at around 445 $cm^{-1}$ the band assigned to the Si-O-Si bending vibrations appeared.

When analyzing the peaks in the bands near 1400 $cm^{-1}$, which are affected by asymmetric stretching bond C-O ($CO_3^{-2}$), we can conclude that these bands were strong in pastes SC and SHC, derived by the presence of calcium or sodium carbonates. The presence of carbonates of different natures was corroborated by the DTG analysis. The formation of the hydrotalcite phase was identified by the presence of the stretching vibration mode of Al-O and Mg-O at around 1050 $cm^{-1}$, and this peak was identified in all the pastes for all three curing ages.

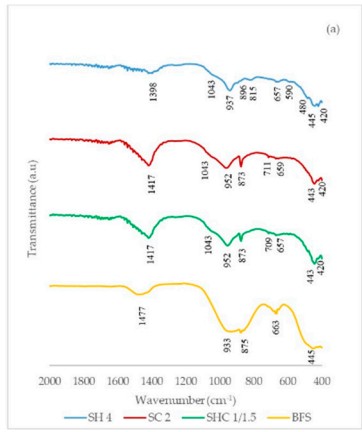
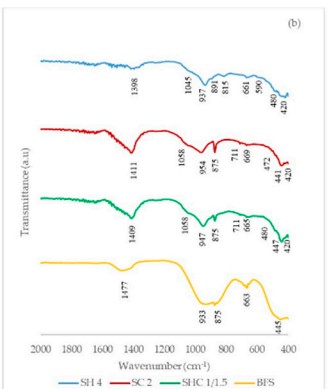
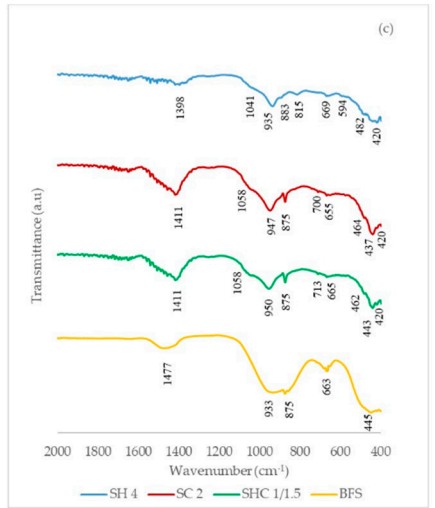

**Figure 8.** FTIR spectra of the pastes cured for: (**a**) 7 days; (**b**) 28 days; (**c**) 90 days.

**Table 4.** Assignment of the bands of the FTIR spectra.

| Wavenumber ($cm^{-1}$) | Affected | Present in the Specters of |
|---|---|---|
| 1398–1477 | $v$ O-C-O ($CO_3^{-2}$) | BFS, SH, SC and SHC |
| 1041–1058 | $v$ Mg–O/Al–O (hydrotalcite) | SH, SC and SHC |
| 933–954 | $v$ Si-O-T (T: Al or Si) (CSH and CASH) | BFS, SH, SC and SHC |
| 873–875 | $v$ C-O ($CO_3^{-2}$) | BFS, SC and SHC |
| 700–713 | $v$ C-O ($CO_3^{-2}$) | SC and SHC |
| 655–671 | $\delta$ Si–O–Si (CASH) | BFS, SH, SC and SHC |
| 590–594 | $\delta$ Si–O–Si/Si–O–Al (NASH) | SH |
| 470–486 | $\delta$ Si–O–Si | SH, SC and SHC |
| 420–447 | $\delta$ Si–O ($SiO_4$ Td) | BFS, SH, SC and SHC |

The $v$ stretching bands and the $\delta$ bending bands.

The band within the 933–954 $cm^{-1}$ range was attributed to the asymmetric stretching vibration of the Si-O-T units (T: Si or A, tetrahedral) of CSH and CASH. This band was found in all the FTIR of the pastes. The bands at 875 and 711 $cm^{-1}$ were visible only in pastes SC and SHC, and these bands were attributed to the carbonated phases. The bands located within the lowest wavenumber range were attributed to the bending bands associated with products such as CSH, CASH or NASH. The FTIR of pastes SC and SHC were similar and only differed slightly from paste SH, especially in the vibration bands corresponding to the carbonated phases.

### 3.4. XRD Analysis

The pastes cured for 28 and 90 days were analyzed by XRD. Figure 9 shows the XRD diffractograms of SH4, SC2 and SHC1/1.5 for both these curing ages. Table 5 represents the abbreviation for the crystalline phases found in XRD, their PDF card number and the pastes that presented this phase.

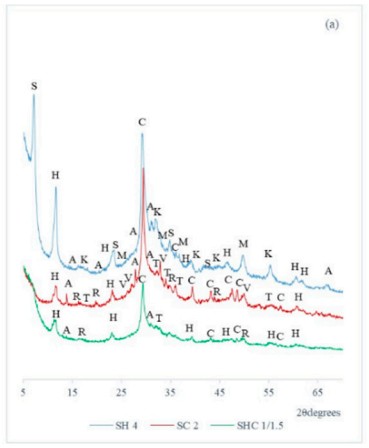 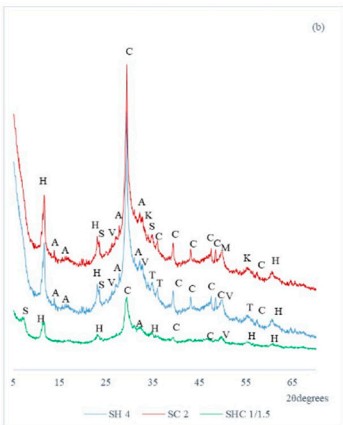

**Figure 9.** XRD spectra of the pastes cured for (**a**) 28 days and (**b**) 90 days.

As seen in Figure 9, calcite is presented in all the pastes. The presence of calcite in the pastes of SC and SHC is logical given the presence of carbonate in the alkaline activator, as corroborated by the FTIR and TG analyses. The presence of calcite in the SH pastes could be due to unreacted slag. The akermanite, monticellite and rankinite peaks may be attributed to the undissolved phases from BFS. Hydrotalcite was present in all the pastes, but the main peak intensity of this phase in the SH paste was higher than that of the other pastes. Moreover, in the SH paste, another peak of the hydrotalcite phase appeared. Tobermorite-11A appeared in the SH pastes at 28 and 90 curing days. There were other differences between pastes that presented carbonates as an activator and the paste with the only hydroxide. Katoite appeared in paste SH, and more intensity appeared signals from the carbonated phases in pastes SC and SHC, with phases vaterite and thermonatrite, and the presence of this type of carbonated phases is common in pastes activated with $Na_2CO_3$ [27,31].

**Table 5.** Assignment of the XRD peaks in pastes.

| Abbreviation | Name | Formula | PDF Card |
|:---:|:---:|:---:|:---:|
| A | Akermanite | $Ca_2MgSi_2O_7$ | 350592 |
| C | Calcite | $CaCO_3$ | 050586 |
| H | Hydrotalcite | $Mg_6Al_2CO_3(OH)_{16}\ 4H_2O$ | 140191 |
| K | Katoite | $Ca_3Al_2(SiO_4)(OH)_8$ | 380368 |
| M | Monticellite | $CaMgSiO_4$ | 350590 |
| R | Rankinite | $Ca_3Si_2O_7$ | 220539 |
| S | Tobermorite-11A | $Ca_5(OH)_2Si_6O_{16}\ 4H_2O$ | 191364 |
| T | Thermonatrite | $Na_2CO_3.H_2O$ | 080558 |
| V | Vaterite | $CaCO_3$ | 240030 |

### 3.5. Economic and Environmental Aspects

The use of chemical reagents to prepare AAM can be taken into account in economic and environmental terms. The selection of a given chemical reagent has repercussions on both the cost and $CO_2$ emissions associated with mortar or concrete preparation.

Regarding economic savings, the selection of NaOH, $Na_2CO_3$, or a mixture of both, influences the cost of chemical reagents. For this study, 712 €/ton of NaOH and 556 €/ton of $Na_2CO_3$ were selected as the cost of reagents (these data were provided by a chemical company). Accordingly, the cost of

the activator per ton of prepared mortar was calculated (Table 6). Obviously, when the amount of activator is increased, the cost per ton rises. For a given $Na_2O\%$, SC mixtures are the most costly because, although the cost of $Na_2CO_3$ is lower, the quantity of $Na_2CO_3$ is larger. All SHC mixtures contain $Na_2O\%$ at 4.96 in relation to the BFS content and 1.11 in relation to the prepared mortar. Hence the cost per ton of mortar increases when the $Na_2CO_3$ proportion does.

**Table 6.** Cost of the activator per mortar, 90-day compressive strength (Rc) and cost of the activator per ton of prepared mortar and megapascal.

| Mixture | Cost of Activator in Mortar (€/ton) | Rc-90 Days (MPa) | Cost per ton and MPa (€/ton × MPa) |
|---|---|---|---|
| SH 3 | 7.7 | 23.3 | 0.33 |
| SH 4 | 10.2 | 25.2 | 0.41 |
| SH 5 | 12.7 | 26.1 | 0.49 |
| SC 1.5 | 7.9 | 41.4 | 0.19 |
| SC 2 | 10.5 | 46.7 | 0.23 |
| SC 2.5 | 13.1 | 51.4 | 0.25 |
| SC 3 | 15.6 | 54.5 | 0.29 |
| SC 3.5 | 18.1 | 52.7 | 0.34 |
| SHC 3/0.5 | 10.3 | 33.0 | 0.31 |
| SHC 2/1 | 10.4 | 35.4 | 0.29 |
| SHC 1/1.5 | 10.4 | 43.1 | 0.24 |

As the strength developed for the prepared mortars was different, the variable cost per ton of mortar and MPa was calculated by taking into account mechanical behavior after 90 curing days (Table 6). It was noticed that the mortars containing $Na_2CO_3$ (SC and SHC series) were less costly (range: 0.19–0.34 €/ton × MPa) than the mortars containing NaOH (SH series, range: 0.33–0.49 €/ton × MPa). This means that selecting $Na_2CO_3$ as an activator is preferable from an economic point of view. The best mixture in relative cost terms per MPa was SC 1.5, which had a value of 0.19 €/ton × MPa. This mortar presented an excellent mechanical performance at 90 curing days (41.4 MPa) for a mixture containing only a little $Na_2CO_3$ (14.2 kg $Na_2CO_3$/ton of prepared mortar).

Regarding $CO_2$ emissions, several data were taken from the bibliography and technical reports. The value of $kgCO_2$ emitted per ton of NaOH was 1.12, as reported by Mellado et al. [12]. $Na_2CO_3$ synthesis is usually carried out by two methods: from natural carbonate ore (usually trona, $Na_3(HCO_3)(CO_3)\cdot 2H_2O$) and by the Solvay process.

In the first case, $Na_2CO_3$ production is carried out by means of trona calcination [39]:

$$2\ (Na_2CO_3NaHCO_3.2H_2O) \rightarrow 3\ Na_2CO_3 + 5\ H_2O + CO_2$$

During this process, carried out at ≈160 °C, two moles of trona produced one mole of $CO_2$ and yielded 3 moles of $Na_2CO_3$. From the stoichiometric calculations, 176.7 kg of $CO_2$ were emitted for each ton of synthesized $Na_2CO_3$. Moreover, fuel consumption must be taken into account: it is reported [40] that 48.7% of the total $CO_2$ emissions related to $Na_2CO_3$ synthesis from trona corresponds to stationary combustion emissions. Thus, for each ton of $Na_2CO_3$, 168.0 kg of $CO_2$ is emitted. Hence, the sum of both contributions is 344.7 $kgCO_2$/ton of $Na_2CO_3$.

In the second case, by the Solvay process, no $CO_2$ emissions were produced when the global chemical reaction was taken into account [41]:

$$2\ NaCl + CaCO_3 \rightarrow Na_2CO_3 + CaCl_2$$

In modern industries, the synthesis of 1 ton of $Na_2CO_3$ by the Solvay process produces 700 kg of $CO_2$ [41] because of the energy used in the several steps followed for synthesis. This value is significantly higher than that corresponding to the process for trona calcination (103% more emitted

$CO_2$). The $CO_2$ values associated with the activator in the mortar, and the quantity of $CO_2$ emitted per ton of mortar and megapascal, are summarized in Table 7.

**Table 7.** Kilograms $CO_2$ emissions from the activator for each mortar per ton of prepared mortar, and per ton of mortar and megapascal, for $Na_2CO_3$ from trona calcination and by the Solvay process.

| Mixture | $Na_2CO_3$ from Trona Calcination | | $Na_2CO_3$ by the Solvay Process | |
|---|---|---|---|---|
| | $kgCO_2$ Activator/ton | $kgCO_2/ton \times MPa$ | $kgCO_2$ Activator/ton | $kgCO_2/ton \times MPa$ |
| SH 3 | 12.1 | 0.52 | 12.1 | 0.52 |
| SH 4 | 16.1 | 0.64 | 16.1 | 0.64 |
| SH 5 | 20.0 | 0.77 | 20.0 | 0.77 |
| SC 1.5 | 4.9 | 0.12 | 9.9 | 0.24 |
| SC 2 | 6.5 | 0.14 | 13.2 | 0.28 |
| SC 2.5 | 8.1 | 0.16 | 16.5 | 0.32 |
| SC 3 | 9.7 | 0.18 | 19. | 0.36 |
| SC 3.5 | 11.2 | 0.21 | 22.8 | 0.43 |
| SHC 3/0.5 | 13.7 | 0.41 | 15.3 | 0.46 |
| SHC 2/1 | 11.3 | 0.32 | 14.6 | 0.41 |
| SHC 1/1.5 | 8.9 | 0.21 | 13.9 | 0.32 |

When the amount of activator increases in the mortar dose, the $CO_2$ associated with this also increases. Once again, the mixtures for the SC and SHC series gave lower $kgCO_2$ values per ton of mortar and the megapascal prepared mortar (range: 0.12–0.41 $kgCO_2$/ton mortar for $Na_2CO_3$ obtained from trona; range: 0.24–0.46 $kgCO_2$/ton mortar by the Solvay process). These values are significantly more advantageous from the environmental point of view than those for the SH series (range: 0.52–0.77 $kgCO_2$/ton mortar). This means that employing $Na_2CO_3$ is preferable to NaOH. Moreover, this reagent is less alkaline, and its handling is safer than it is for NaOH. Thus, selecting $Na_2CO_3$ would be the best choice.

## 4. Conclusions

This paper investigated the alkali activation of slag to be used mainly in developing countries. To do so, various more economical and environmentally friendly contents, such as $Na_2CO_3$, were used as activators and compared to the use of NaOH. The highlighted conclusions from this study are:

a. All the mortars activated with NaOH (SH series) for 28 curing days had compressive strength values of around 20 MPa, while all the mortars activated with $Na_2CO_3$ (SC series) had values over 40 MPa at the same curing ages. The mortars with the two activators (SHC series) had intermediate values, between 27 and 40 MPa. All the values obtained by the SC series offered optimum use for nonstructural elements in, for example, blocks, tiles or pavements.

b. Alkali-activated slag with 3.5 M of sodium carbonate led to higher compressive strengths: $56.5 \pm 1.83$ MPa and $9.3 \pm 0.4$ MPa tensile strength at 28 curing days. This mixture can be used for structural elements.

c. Characterization studies by TG/DTG, FTIR, and XRD confirmed the presence of CSH, hydrotalcite, CASH and carbonated phases, such as thermonatrite or calcite/vaterite.

d. The environmental and economic studies showed that the samples activated with $Na_2CO_3$ yielded the best results. When comparing the SC with the SH series, reductions over 40% in cost per ton and MPa, and higher than 50% of $kgCO_2$ emitted per ton and MPa, were obtained.

**Author Contributions:** Conceptualization, N.B., J.M.M. and E.G.; methodology, M.V.B., J.M.M. and J.P.; formal analysis, A.F., L.S. and M.V.B.; investigation, A.F., L.S., J.P., M.V.B., N.B.; E.G. and J.M.M.; resources, N.B., E.G. and J.M.M.; writing—original draft preparation, N.B.; J.M.M.; L.S. and A.F.; writing—review and editing, J.P. and M.V.B.; visualization, A.F., J.M.M., M.V.B.; J.P., supervision, J.M.M., M.V.B. and J.P.; project administration, J.M.M.; funding acquisition, J.M.M. All authors have read and agreed to the published version of the manuscript.

**Funding:** We would also like to thank the Spanish Government MINECO/FEDER (ECOSOST project RTI-2018-097612-B-C21) for supporting this research.

**Acknowledgments:** This investigation was done during the postdoctoral scholarship of Erasmus Mundus EMMAG program second action. For this, the authors would like to thanks Erasmus Mundus and the EMMAG program team and especially Ester Durá and Elena Taulet in OAI of UPV for their assistance and help. We would also like to thank the UPV Electronic Microscopy Service and DGRSDT.

**Conflicts of Interest:** The authors declare no conflict of interest.

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
