# Peer review of "Formulation of Alkali-Activated Slag Binder Destined for Use in Developing Countries"

_applsci, doi:10.3390/app10249088_

Round 1

Reviewer 1 Report

The paper is well written. Minor revision is required before it can be accepted for publication. My comments are shown below. 

Comment 1: The text in the figures are too small to read.

Comment 2: Line 112, for the chemical analysis results for BFS, SO3 is not likely to exist. It is quite misleading if the authors show it like this. 

Comment 3: Line 125, the authors claim that the phase of CaCO3 could be observed from the XRD pattern. However, this is not likely to be the case. Because there is only one characteristic peak for CaCO3, and in the BF slag it is not likely that CaCO3 could exist. 

Comment 4: Line 205, "contianing" ? check the spelling. 

Comment 5: Lines 280-284, more explanation for the results from Figure 7 is required. What are the differences concerning different curing days?

Author Response

Reviewer #1:

Comments to the authors:

Comment 1: The text in the figures are too small to read

We agree with the reviewer and we have changed the font size of Figures

Comment 2: Line 114, for the chemical analysis results for BFS, SO3 is not likely to exist. It is quite misleading if the authors show it like this

The common way to express the results obtained by FRX analysis is in the form of oxides. This composition express that there are components with this element (S) not that the SO3 is present in the BFS.

Comment 3: Line 127, the authors claim that the phase CaCO3 could be observed from the XRD pattern. However, this is not likely to be the case. Because there is only one characteristic peak for CaCO3, and in the BF slag it is not likely that CaCO3 could exist.

The mainly amorphous character of the sample makes the crystalline peaks unclear, the existence of carbonates in the BFS is also found in other analysed papers. For example in the reference: Puertas, F.; Torres-Carrasco, M. Use of glass waste as an activator in the preparation of alkali-activated slag. Mechanical strength and paste characterisation. Cem. Concr. Res. 2014, 57, 95–104.

Comment 4: Line 202, “containing”? check the spelling

The word has been corrected.

Comment 5: Lines 281-283, more explanation for the results form Figure 7 is required. What are the differences concerning different curing days?

A new text has been added indicating the main changes occurring with the curing age: “The mass loss in the interval 50-300ºC presented a great evolution between 7 and 28 curing days; this data confirms that the most developed products within this time interval are CSH and CASH gel”

Reviewer 2 Report

The article “formulation of alkali-activated slag binder destined for use in developing countries” reports the formulation of AAM using activators such as NaOH, Na2CO3 and the combination of these two bases. It is also reported the characterization of the AAM using different techniques and their economic and environment benefits were presented. The paper has a couple of mistakes, which should be addressed before the manuscript will be ready for the acceptance for publication to section: Environmental and Sustainable Science and Technology. The comments are given as below:

Abstract: The authors refer “Three activators were used: NaOH, Na2CO3 and a mixture of both” but I suggest two activators were used NaOH and Na2CO3, a mixture of both was also used/tested.

It is necessary correct some mistakes along the text: put “et al” in italic format; correct “ºC” for “°C”.

Page 2, line 72. Please provide the abbreviation meaning of “C-S-H” and “C-A-S-H”.

Page 2, line 78-79. The explanation for the use of Na2CO3 as an environment-friendly and economic alternative is not clear. The authors should be providing a better explanation.

Page 6, line 197. The authors report “…strength slightly decreased in all the mortars after 90 curing days”. The results show that SC3 strength slightly increased. Please revised.

Page 6, line 205. Please correct “contianing”.

Page 12, table 5. Please verify the formula of hydrotalcite. The general formula is Mg6Al2(CO3) (OH)16≅ 4H2O.

Page 14, Na2CO3 production equation, trona calcination. Correct the equation and use chemical conversion arrows instead of equal. See reference 38.

Page 14, Na2CO3 production equation, Solvay process. Please use chemical conversion arrows instead of equal.

The authors could also introduce these references in the manuscript:

Ahmad M. R.; Chen B.; Shah S. F. A. Influence of different admixtures on the mechanical and durability properties of one-part alkali-activated mortars, Construction and Building Materials, 265, 2020, 120320.

Awoyera P.; Adesina A.  A critical review on application of alkali activated slag as a sustainable composite binder, Case Studies in Construction Materials, 11, 2019, e00268.

Author Response

Reviewer #2:

Comment 1: Abstract: The authors refer “Three activators were used: NaOH, Na2CO3 and a mixture of both”, but I suggest two activators were used NaOH and Na2CO3, a mixture of both was also used/tested.

The text has been corrected as indicated by the reviewer

Comment 2: it is necessary correct some mistakes along the text: put “et al” in italic format, correct ºC for ºC

The text has been corrected as indicated by the reviewer

Comment 3: Page 2, line 72. Please provide the abbreviation meaning of “C-S-H and C-A-S-H”

The text has been unified by putting CSH and CASH and commenting on the full name, calcium silicate hydrate and calcium silico-aluminate hydrate

Comment 4: Page 2, line 81-82. The explanation for the use of Na2CO3 as an environment-friendly and economic alternative is not clear. The authors should be proving a better explanation

This statement refers to a consulted article (17) and is subsequently substantiated by the economic and environmental study with the research data. The authors added the reference proposed by the reviewer and the text was modified: “Awoyera and Andesina [18] made a review on application of alkali activated slag and also consider Na2CO3 the most appropriate activator”. The new reference occupies the 18th position.

Comment 5: Page 6, line 194. The authors report “…strength slightly decreased in all the mortars after 90 curing days”. The results show that SC3 strength slightly increased. Please revised.

The text was modified “After 28 curing days, all the values were similar and strength slightly decreased in all the mortars after 90 curing days except for the mortar SC3”

Comment 6: line 202

The word has been corrected.

Comment 7: page 12 table 5, please verify the formula of hydrotalcite. The general formula is Mg6Al2(CO3).(OH)16.4H2O

The formula has been corrected.

Comment 8: Page 14 Na2CO3 production equation, trona calcination. Correct the equation and use chemical conversion arrows instead of equal. See reference 38

The text has been corrected

Comment 9: Na2CO3 production equation, Solvay process. Please use chemical conversion arrows instead of equal

The text has been corrected

Comment 10: The authors could also introduce these reference in the manuscript

The new reference 18 is one of the proposals by reviewer 2.